# EVOLUTIONARY FEDERATED LEARNING USING PARTICLE SWARM OPTIMIZATION

## ABSTRACT

Efficient communication is a key challenge in federated learning, where multiple clients contribute to a shared model. To address this issue, reducing local computation is an effective solution. This paper proposes an innovative federated learning algorithm that utilizes Particle Swarm Optimization, a powerful evolutionary algorithm, to minimize the computational demands on federated learning clients. Our results show that this algorithm results in significant enhancements in accuracy and faster convergence of loss compared to traditional federated learning methods.

## 1    INTRODUCTION

In recent years, Federated Learning has emerged as a promising approach to privacy-preserving machine learning. This method allows clients to train models locally while keeping their sensitive data securely on their own devices, as opposed to traditional centralized learning where clients send their data to a central server (Konečný et al., 2016). In federated learning, communication efficiency is a critical consideration. To improve the communication bottleneck, reducing the computational demands on clients is a crucial step. Evolutionary algorithms, with their ability to approximate the minima of an objective function, present a potential solution (Vikhar, 2016). Compared to deterministic gradient-based optimization algorithms, evolutionary algorithms are computationally less demanding and do not guarantee convergence (Hamed et al., 2010).

**Related Work:**  Previous works have explored different approaches to reduce the communication cost in Federated Learning. For example, structured updates and sketched updates can reduce the communication cost in FedAvg by two orders of magnitude (Konečný et al., 2016). Additionally, lossy compression and Federated Dropout have been proposed to reduce communication costs in FedAvg without degrading the quality of the final model (Wen et al., 2021).

Moreover, compression objectives such as gradient compression, model broadcast compression, and local computation reduction have been studied in the literature (Kairouz et al., 2019). Most work focuses on gradient compression because it takes more time to upload data than to receive data on most edge devices. In contrast, our approach focuses on reducing local computation, which can be combined with model broadcast compression and compression of model update compression.

Other studies have explored the use of evolutionary algorithms to optimize the federated learning process. For example, the Ditto algorithm trains a different model on each client to ensure the model performs well on every device (Li et al., 2020). Similarly, we focus on our algorithm's performance on client devices and pursue fairness by reducing the local computation required from any client participating in each round of federated training. Lastly, FedGKT (He et al., 2020) posits federated learning as a group knowledge transfer problem, where small CNNs on client devices receive knowledge from a large CNN stored on the server. In FedGKT, knowledge transfer reduces the local computation required from any client participating in a round of federated training.

In this paper, we propose the use of Particle Swarm Optimization (PSO), a powerful and efficient evolutionary algorithm, to enhance federated learning. PSO is well-known for its low memory and computational requirements (Poli et al., 1995). By integrating PSO into federated learning, clients can train their models with fewer computational resources, leading to improved communication efficiency and faster convergence of the shared model.

## 2 EXPERIMENTAL SETUP

**Centralized Training**    In this experiment, we adopted a knowledge transfer approach to reduce the computational load on client devices. We utilized a pre-trained MobileNetV2 model, freezing its first five layers and transferring the weights to a new architecture implemented in Keras (He et al., 2020). To the transferred model, we added several layers, including a Flatten layer, a Dense layer, a ReLU activation function, a Dropout layer, and a final Dense layer with ten units and a softmax activation. The model was trained using the Adam optimizer and a categorical cross-entropy loss function, with the accuracy evaluated at each location in the hyperparameter search space. Our dataset consisted of 60,000 MNIST training examples and 10,000 test examples, and the experiments were run on an Intel Iris Plus Graphics 1536 MB GPU.

**Federated Setting**    In this setup, we simulated federated learning in a client-server architecture using TensorFlow Federated and Flower as our training frameworks (Bonawitz et al., 2020; Beutel et al., 2020). We trained on the EMNIST and FEMNIST datasets (Cohen et al., 2017; Caldas et al., 2018) and aggregated the weights on the server side using stochastic weight averaging (Izmailov et al., 2018).

## 3 RESULTS

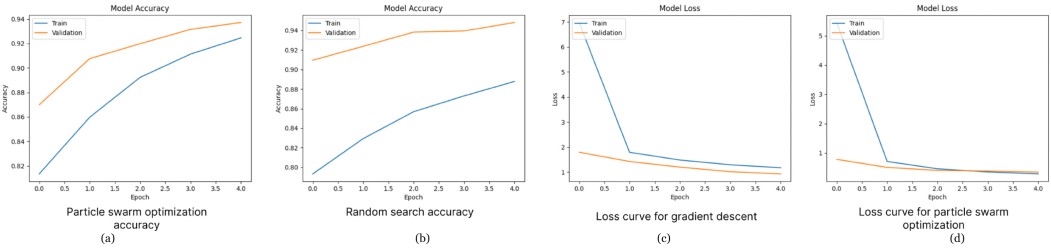

Figure 1: Comparison of model accuracy and loss over time between training (blue) and validation sets (orange) for particle swarm optimization, Fig. 1 (a) & (c), and random search, Fig. 1 (b) & (d).

As shown in Fig. 1, an early decline in the loss curve for particle swarm optimization is observed as compared to gradient descent. Furthermore, as observed in Table 1, particle swarm optimization performed better than gradient descent when it comes to accuracy, although taking a longer time for the same number of epochs.

| Optimization | Time | Accuracy |
|---|---|---|
| Particle swarm | 86.1671 sec. | 0.9624 |
| Gradient descent | 52.6553 sec. | 0.6814 |
| Random search | 44.4513 sec. | 0.9501 |

Table 1: Comparison of optimization time and accuracy for different approaches.

## 4 DISCUSSION

The results of our study demonstrate the potential of particle swarm optimization to reduce local computation in federated learning, thus improving communication efficiency and reducing the rounds of communication needed to train a shared model. The reduced local computation achieved through particle swarm optimization has the potential to make the training process faster and more efficient, as fewer communication rounds are needed to train the shared model. In addition, by reducing the computation load on client devices, this approach can help to overcome some of the challenges posed by resource-constrained clients. There are several areas of future work that can build on the results of this study. Firstly, we plan to replicate the experiments through client-server simulations to validate the results. Additionally, our proposed method can be further improved by incorporating techniques such as accelerated particle swarm optimization (Gandomi et al., 2013).

URM STATEMENT

The authors acknowledge that all the authors of this work meet the URM criteria of ICLR 2023 Tiny Papers Track.

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
