# OpenReview forum: "Evolutionary Federated Learning Using Particle Swarm Optimization"
_ICLR.cc/2023/TinyPapers — Submitted to Tiny Papers @ ICLR 2023_

### Official Review · Reviewer_8YyB · 2023-04-01

**Confidence:** 2

**Summary Of Contributions:**

This paper proposes to use Particle Swarm Optimization (PSO) for federated learning. The paper shows improved accuracy (but longer computation times) in comparison to Gradient Descent and Random search on MNIST-based data, in a Federated Learning setting.

**Rating:**

Great Start (GS): a submission which meets some of the reviewing criteria but has room for improvement

**Strengths And Weaknesses:**

### Strengths
- Good motivation.
- Discussion of related work appears solid, although I am not very familiar with all the literature on Federated Learning myself.
- Good start to the experiments.

### Weaknesses
1. I do not entirely follow the reasoning that "To improve the communication bottleneck, reducing the computational demands on clients is a crucial step." Maybe I'm just missing something obvious, but I don't see a direct relation between computational demands on clients, and communication costs. In fact, I'd assume that in some cases you could **reduce** communication costs by **increasing** per-client computation (for example, if per-client computation is used for more effective compression or summarisation of data or results).
2. It appears to me that there is a disconnect between the results and the claims. Again, maybe I'm missing something obvious. But I see various claims (in abstract, introduction, discussion) about **faster** convergence, **fewer** computational resources, and so on, and then in the results (Table 1) I see that **more** computation time was used. Maybe for fewer epochs, yes. But isn't computation time the most important indicator of speed and computation resources? More so than epochs?
3. I would appreciate more clarity in the presentation of the results. Subfigures (a) and (b) of Figure 1 should have equal ranges for their y-axes, and the same counts for (c) and (d). They are already similar, but not exactly equal. I am also unsure about what some of the subfigures actually show, since there is a mismatch between text in the figures, and text in the caption. The caption says that subfigures (a) and (c) are for PSO, but the text on the subfigures themselves suggests that (a) and (d) are for PSO. The caption says that (b) and (d) are for Random Search, but the text inside the figures says that only (b) is for Random search, with (d) being for PSO, and (c) being for Gradient Descent.

**Suggested Changes:**

I would recommend making changes to clarify the points I listed as Weaknesses in my review above. I think point 3) is the most important one, to clarify and clean up the presentation of the results. This is probably followed by 2), and finally 1) in terms of importance. These last two points I'm somewhat less sure about: maybe I'm just confused due to my lack of familiarity with Federated Learning literature. However, it is important for the work to be as self-contained as possible, and ideally it should not confuse even readers who are not familiar with all related work.

---

### Official Review · Reviewer_3C44 · 2023-04-04

**Confidence:** 2

**Summary Of Contributions:**

This paper proposes a federated learning algorithm which uses Particle Swarm Optimization (PSO). It aims to reduce the computation load on client devices, faster loss convergence (which however seems to be inconsistent with the results) and higher accuracy than traditional FL techniques. The authors run experiments on EMNIST and FEMNIST data and compare the PSO based FL with random search and gradient descent.

**Rating:**

Clear, Correct, and Reproducible (CCR): a submission which meets the reviewing criteria

**Strengths And Weaknesses:**

Strengths:
1. The paper is well written and tackles an important problem of reducing communication in federated learning.
2. The experimental setup is clearly explained.
3. The relevant work section is well-researched.

Weakness:
1. The paper claims that incorporating PSO with federated learning results in faster convergence of loss than traditional federated learning algorithms. However, the number of epochs for PSO and gradient descent remains the same, and the time taken for the same number of epochs increases for PSO. It would be interesting to see a thorough analysis on this.
2. In Fig. 1, the (a) and (d) figures are captioned to be for PSO based FL. However, the description for Fig. 1 mentions (a) and (c) for PSO based FL. The description and the figures are ambiguous on (b) and (c) as well, with figures mentioning that (b) and (c) represent random search accuracy and gradient descent loss while the description mentions something else.

**Suggested Changes:**

Same as weaknesses

---

### Comment · Area_Chair_3BoR · 2023-06-08
**Archival Criterion Check**

The paper has not been revised.

---

### Meta-Review · Area_Chair_3BoR · 2023-04-07

**Recommendation:** Invite to revise
**Confidence:** 3

**Metareview:**

Both reviewers raise questions regarding _clarity_ and _correctness_ of the paper.

After reading the paper, I find myself in total agreement with almost all of reviewer `8YyB`'s points and wouldn't have much to add to them. In my view, reviewer `8YyB`'s comments are fair and also encapsulate most of reviewer `3C44`'s points as well.

In addition, however, I would also like to point out that I am unsure how the central thesis of the paper (applying Particle Swarm Optimization (PSO) in federated learning) was actually used (e.g., to optimize what?). I'm also unclear as to how PSO performed only as well as random search when random search takes half as much computation but with only a ~1% inferior performance. It is also very surprising that both PSO and random search perform far superior to gradient descent; why this is should be addressed as well.


**Summary:**

The paper proposes to use Particle Swarm Optimization to reduce communication costs in federated learning. The paper is well-motivated and easy to read, but there are issues with the correctness and clarity aspects of it.

**Reason For Not Giving A Higher Recommendation:**

Serious issues raised by both reviewers that must be addressed.

**Reason For Not Giving A Lower Recommendation:**

N/A

---

### Decision · Program_Chairs · 2023-04-08

No revision received; not invited to archive